# HIV-1 RNA in extracellular vesicles is associated with neurocognitive outcomes

Catherine DeMarino [1,4], Julia Denniss[1,4], Maria Cowen[1], Gina Norato[2], Devon K. Dietrich[1], Lisa Henderson[1], Elyse Gollomp[3], Joseph Snow[3], Darshan Pandya[1], Bryan Smith[1] & Avindra Nath [1] ✉

Human immunodeficiency virus type-1 (HIV-1) is responsible for significant mortality and morbidity worldwide. Despite complete control of viral replication with antiretrovirals, cells with integrated HIV-1 provirus can produce viral transcripts. In a cross-sectional study of 84 HIV+ individuals of whom 43 were followed longitudinally, we found that HIV-1 RNAs are present in extracellular vesicles (EVs) derived from cerebrospinal fluid and serum of all individuals. We used seven digital droplet polymerase chain reaction assays to evaluate the transcriptional status of the latent reservoir. EV-associated viral RNA was more abundant in the CSF and correlated with neurocognitive dysfunction in both, the cross-sectional and longitudinal studies. Sequencing studies suggested compartmentalization of defective viral transcripts in the serum and CSF. These findings suggest previous studies have underestimated the viral burden and there is a significant relationship between latent viral transcription and CNS complications of long-term disease despite the adequate use of antiretrovirals.

Human immunodeficiency virus type-1 (HIV-1), the causative agent of acquired immunodeficiency syndrome (AIDS), is responsible for significant mortality and morbidity worldwide since its discovery in 1981[1]. As of 2021, roughly 28.7 million of the ~38.4 million people living with AIDS globally were receiving combined antiretroviral therapy (cART)[1]. Despite highly effective antiretroviral drugs, cells with chromosomally integrated HIV-1 provirus are present in the central nervous system (CNS) and can produce viral transcripts[2–7]. Viral RNA can also be detected in cells derived from several regions of the brain, including the frontal cortex, and the cerebrospinal fluid (CSF)[4]. More recently, we found that HIV-1 transcripts are present in extracellular vesicles (EVs), including exosomes, derived from primary cell culture supernatant, cerebrospinal fluid and plasma[8–11], which has been confirmed by other studies[12–14].

EVs are small, membrane-bound particles released from numerous cell types that function in cell-to-cell communication. They range in size from 30 nm to 1 μm in diameter and vary in their mechanisms of release and associated cargos. There is an intimate relationship between viruses and EVs[15]. In cART treated individuals, even if fully replicating viral particles cannot be formed, viral RNA and protein can be packaged in EVs and transported. Thus, the content of the EVs reflect the cells of origin. This is particularly important for studying brain viral reservoirs in persons living with HIV (PLWH) since brain derived EVs can be isolated from CSF[16]. A thorough understanding of these viral reservoirs is necessary not only for cure strategies but for understanding the role of viral persistence in the pathophysiology of HIV-1 associated neurocognitive disorders (HAND), a phenotype which is exhibited in 20–50% of PLWH[17]. In this work, we investigated viral persistence in the CNS and peripheral blood viral reservoirs using a series of digital droplet polymerase chain reaction assays and sequencing in a cohort of well characterized PLWH who were well controlled on antiretrovirals and monitored annually as part of a

[1]Section for Infections of the Nervous System, National Institute of Neurological Disorders and Stroke, National Institutes of Health, Bethesda, MD, USA. [2]Office of the Clinical Director, National Institute of Neurological Disorders and Stroke, National Institutes of Health, Bethesda, MD, USA. [3]Office of the Clinical Director, National Institute of Mental Health, National Institutes of Health, Bethesda, MD, USA. [4]These authors contributed equally: Catherine DeMarino, Julia Denniss. ✉e-mail: natha@ninds.nih.gov

natural history study. Overall, these findings suggest that the persistence of viral transcription in virologically suppressed patients may play a role in mediating HAND phenotypes, opening a new approach to addressing neurocognitive disorders in HIV-1 infected individuals.

## Results

Numerous studies have shown the presence of viral transcription despite effective antiretroviral therapy, suggesting a transcriptionally active latent reservoir. We have adapted previously published reverse transcriptase digital droplet polymerase chain reaction (RT-ddPCR) assays for several different regions of HIV-1[18] to evaluate the transcriptional profile of the virus in each reservoir using EV RNA (Fig. 1a). These include assays for transcriptional readthrough (U3-U5), transcription initiation/elongation block (TAR), elongation of RNA transcripts beyond the 5′ LTR (R-U5/Gag), elongation beyond Gag (Pol), distal elongation (Nef), completion of transcription (polyA), and splicing (Tat-Rev) (Fig. 1b). A cross-sectional subset (Table 1 and Supplementary Data Table 1) and a longitudinal subset (Supplementary Data Table 2) of a larger natural history study cohort were analyzed using the described assays.

### Distinct HIV-1 transcriptional profiles in blood and CSF

Analysis of cross-sectional, visit-matched CSF and serum samples from 84 individuals showed that viral RNA was present in higher copy number in the CSF compared to serum EVs (Fig. 1c–i). The copy numbers were significantly higher in 6 of 7 ddPCR assays with a lack of statistical significance in PolyA copy number. These trends did not correlate with EV numbers in each of the compartments (Supplementary Data Fig. 1). Notably, the CSF contained on average 2 log fewer EVs as compared to the serum in every individual. Different proportions of individuals had HIV transcripts in both CSF and serum, exclusively in CSF, exclusively in serum, and undetectable in both compartments depending on the viral RNA that was analyzed (Fig. 1j). However, there was no individual in whom none of the viral transcripts could be detected. Taken together, these results suggest that the CSF is a transcriptionally active reservoir that is distinct from the peripheral blood.

### Viral EV RNA is associated with neurocognitive dysfunction

The relationship between detectability of each viral transcript in the CSF or serum and neurocognitive functioning (domain scores) was investigated using a Wilcoxon Rank-Sum test. All significant correlations were with CSF transcripts (Fig. 2). Detectability of CSF TAR correlated with a lower summative T-score, indicative of a reduction in general neurocognitive functioning. The detectability of the Long LTR amplicon correlated with a higher global deficit score (GDS; Fig. 2b), a lower summative T-score and decreased performance in Executive Functioning, Attention and Working Memory, Learning, and Memory domains (Fig. 2c–g). Interestingly, the detection of the fully spliced Tat-Rev transcript was associated with higher scores in the Psychomotor domain (Fig. 2h). The most prevalent comorbidities (hepatitis C, diabetes, depression, hyperlipidemia, and anxiety) were found to have a significant association with lower levels of viral RNA (Supplementary Fig. 2a–e), the reasons for which are not clear. Medications used to treat these conditions are not known to have antiviral effects, although many are proposed to have neuroprotective properties which may be a contributory factor[19]. Hyperlipidemia was associated with HAND diagnosis (Supplementary Fig. 2d), anxiety was associated with higher GDS (Supplementary Fig. 2e), and hypertension was associated with age, duration of HIV infection and duration of ART (Supplementary Fig. 2f).

Cross-sectional data was also analyzed using Spearman correlations to visualize the relationships between HIV-1 EV RNA and neuropsychological data collected at the time of lumbar puncture and blood draw. Correlation analysis showed that the copy number of Long LTR in CSF correlated with lower overall T-Scores, a composite measure of cognitive deficits (Fig. 3a). CSF Long LTR further correlated with domain scores for Executive Functioning and Working Memory. As expected, in general, transcripts were positively correlated with other transcripts within the same compartment, except for CSF polyA which negatively correlated with CSF Pol and Tat-rev transcripts. Additionally, CSF TAR negatively correlated with serum read through and CSF Nef negatively correlated with serum Long LTR. This suggests the possibility that suppression of polyA transcripts by ART may result in increased production of the spliced products. The correlation coefficients and *p* values for each test are summarized in Supplementary Data Table 2. Linear regression analysis (Fig. 3b–d) adjusted for covariates including duration of HIV infection, duration of infection that was untreated, duration of antiretroviral therapy, nadir CD4 counts, and viral load confirmed these relationships. Each transcript was further investigated using Wilcoxon Rank-Sum test to investigate differences in viral transcripts between males and females. None of the transcripts differed significantly between males and females. These findings demonstrate that there is persistent viral transcription despite adequate control of viral replication in HIV-1 infected individuals and correlates with neurocognitive deficits.

The effect of CNS penetration of various antiretroviral regimens was assessed by assigning CNS penetration effectiveness (CPE) scores for each antiretroviral drug. CPE scores are assigned based on pharmacokinetics, pharmacodynamics, and physiochemical properties of each drug[20–22]. The CPE scores for each individual were summed and the participants were binned into low CPE (total CPE less than or equal to 8) or high CPE (total CPE greater than or equal to 9) categories. A Wilcoxon Rank-Sum test was used to determine significant differences between groups. There were no significant differences in CSF RNA transcripts between low CPE and high CPE individuals (Supplementary Data Fig. 3), likely due to the lack of a transcription inhibitor in ART regimens. In the serum, readthrough and polyA transcripts were significantly lower in those with high CPE scores.

### Changes in viral RNA copy number correlate with changes in neurocognitive function

Given the relationship between CSF Long LTR and poorer scores on neurocognitive assessments, a sub cohort of 43 HIV-1 infected individuals with two or more visits (23 individuals with 2 visits, 19 individuals with 3 visits, and 1 individual with 4 visits) was analyzed for the presence of TAR and Long LTR RNAs in both the CSF and serum to determine longitudinal correlations. Individual trajectories of all longitudinal transcripts are shown in Supplementary Data Fig. 4. The number of transcripts does not follow a discernible trend over time except for CSF TAR which may increase over time. A linear model was fit to each individual on each metric for all available data to assess pairwise spearman correlations across all variables (Fig. 4a). The pairwise correlations and associated *p* values are summarized in Supplementary Data Table 3. An increase in copy number of Long LTR in the CSF over time was associated with a decline in Executive Functioning domain scores (Fig. 4b) and an increase in TAR copy number in the serum over time correlated with an increase in the individuals Global Deficit Score, which indicates a reduction in overall neurocognitive function.

### Sequencing suggests viral reservoir compartment crosstalk

The Long LTR amplicon was cloned and sequenced from 10 individuals' visit-matched CSF and serum. The Long LTR amplicon contains the polyadenylation (polyA) signal (purple), primer activation site (blue), tRNA stabilizing motif (yellow), anticodon binding motif (red), and a portion of the primer binding site (PBS)[23] (Supplementary Data Fig. 5). Overall, the polyA signal and PBS were largely conserved with very few mutations in the majority of sequences for each individual in the CSF and serum (Supplementary Data Fig. 6). However, many

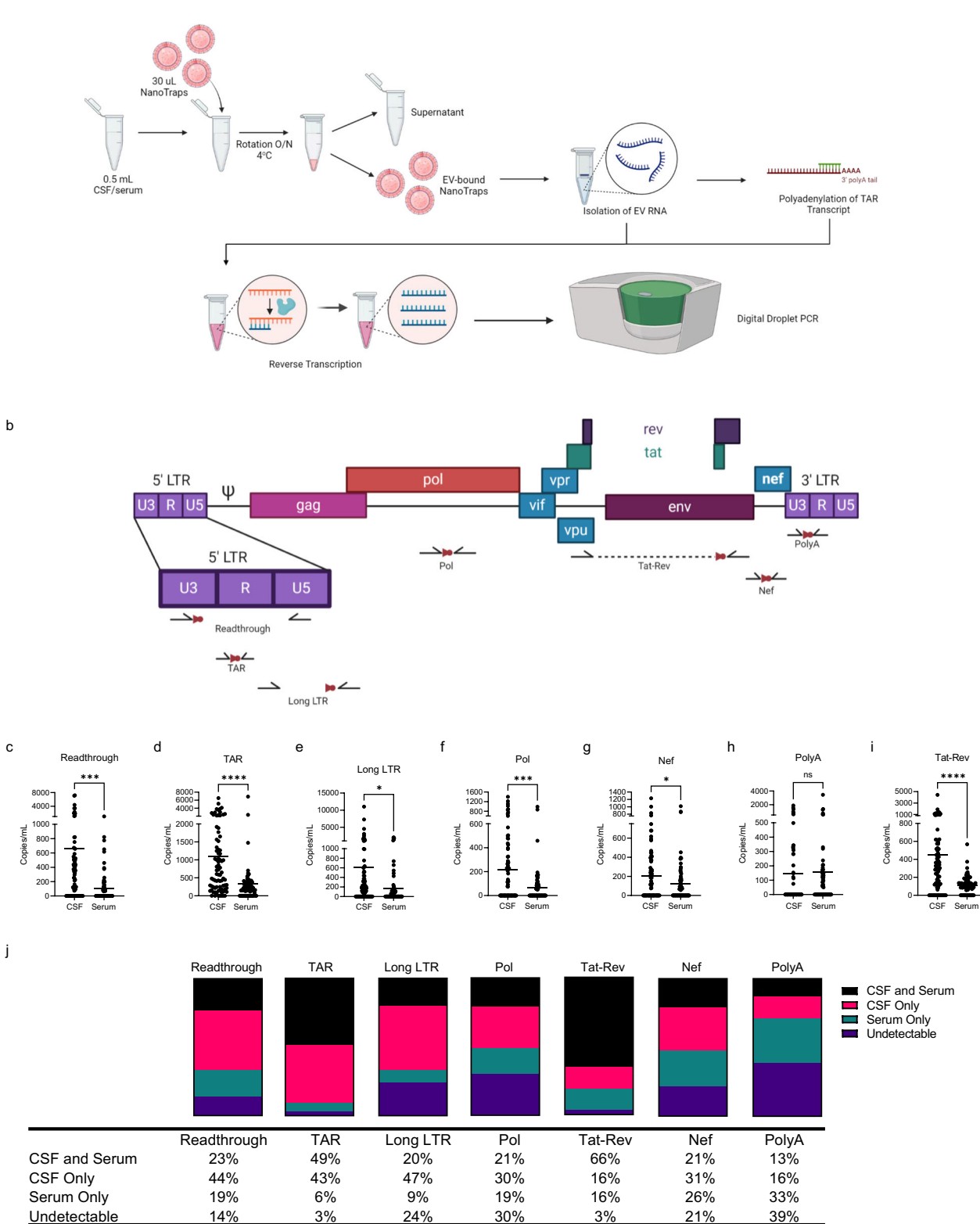

**Fig. 1 | EV-associated HIV RNA is detectable at higher levels in the CSF compared to serum.** Sample processing workflow (**a**) and primer location (**b**) for all samples and ddPCR assays for CSF and serum. Created with Biorender.com. EV RNA from matched CSF and serum from 84 individuals with HIV were analyzed using ddPCR in technical triplicate for (**c**) readthrough transcription, $p = 0.011$; (**d**) TAR, $p = <0.000009$; (**e**) Long LTR, $p = 0.012$; (**f**) Pol, $p = 0.0003$; (**g**) Nef, $p = 0.0314$; (**h**) PolyA; and (**i**) Tat-Rev, $p = <0.000006$. RNA was compared using a two-sided unpaired Student's $t$ test using an average of 3 technical replicates for each of matched CSF and serum from $n = 84$ individuals. **j** Compartmental RNA detectability is presented as percentages of the total 84 tested samples. CSF cerebrospinal fluid, EV extracellular vesicle, TAR transactivation response element, LTR long terminal repeat. Source data are provided as a Source Data file.

**Table 1 | Cross-section demographics**

| Type | Variable | HIV+ (n = 84) |
|---|---|---|
| Demographics | Age—mean years | 57.1 (8.64) |
| | Sex, female—n (%) | 28 (33.3%) |
| | Race, White—n (%) | 38 (45.2%) |
| | Race, Black—n (%) | 44 (52.4%) |
| | Education, >12 years—n (%) | 55 (65.5%) |
| | Education—mean years | 14.1 (3.0) |
| HIV disease | Duration HIV infection—mean years | 21.8 (9.9) |
| | Duration untreated HIV—mean years | 5.3 (7.5) |
| | Duration treated ART—mean years | 16.5 (9) |
| | CD4 count[a]—median (IQR) | 627 (397) |
| | Nadir CD4[b] (n = **80**)—median (IQR) | 184 (173) |
| Regimen—n (%) | Ever on Zidovudine (AZT) | 23 (27.4%) |
| | Ever on Efavirenz (EFV) | 26 (31.0%) |
| | Ever on a "D-drug" (n = **70**) | 7 (10%) |
| | Currently on AZT | 0 (0%) |
| | Currently on EFV | 5 (6.0%) |
| | Currently on a "D-drug" | 0 (0%) |
| | Currently on a protease inhibitor | 14 (16.7%) |
| | Currently on an integrase inhibitor | 64 (76.2%) |
| | Currently on an entry inhibitor | 2 (2.3%) |

Numbers in parentheses represent standard deviations. Numbers in bold indicate the number of individuals out of the 84 individuals studied for which this information was available.
*D-drug* dideoxynucleoside analogues (stavudine [d4T], didanosine [ddI], zalcitabine [ddC]).
[a]Measured within 6 months of neuropsychological testing.
[b]If values were recorded as <100, <200, or >200 cells/µl, 99, 199, and 201 were used, respectively, in analyses.

detected sequences had large deletions which spanned the primer activation site, tRNA stabilizing motif, and anticodon binding motif (shown in gray). For some individuals, the large deletions were found in the CSF and serum, while in others the deletions were primarily found within the CSF (#9, #33, and #56). The majority of individuals (7/10) had one primarily intact sequence which was found in both the CSF and serum alongside deleted sequences that were unique to each compartment. Interestingly, three of the 10 individuals (#36, #42, and #56) had complete compartmentalization of Long LTR sequences, and two individuals (#6 and #65) had two sequences found within the CSF and serum, one intact and one deleted, potentially suggesting reseeding of the one compartment by the other.

A portion of the HIV envelope, which spans the C2-V3-C3 regions, was also cloned, and sequenced out of visited-matched CSF and serum from 10 individuals (Supplementary Data Fig. 7). This amplicon contained amino acids critical for the R5 phenotype (purple), CD4 binding (blue), glycosylation (green), and the neutralizing domain (red)[24,25]. Similar to Long LTR, the majority of individuals had a predominately intact sequence which could be detected in the CSF and serum. However, a large proportion of the serum sequences were deleted in the V3-C3 region, which contains the cytotoxic T-lymphocyte epitope, several glycosylation sites, three amino acids involved in the R5 phenotype, and amino acids essential for CD4 interaction in the C3 region.

## Discussion

This is the largest study to date assessing HIV RNA in matched CSF and serum EVs from virally suppressed PLWH on antiretrovirals. HIV RNA was detected in the CSF and serum from all individuals. The copy numbers of HIV RNA were higher in the CSF compared to the serum, suggesting that the CNS reservoir is highly transcriptionally active. The viral transcripts in the CSF and serum were discordant in this study, suggesting compartmentalization of the virus in these two separate reservoirs. This discordance was largely in the defective viral sequences, suggesting that the selective pressures on the virus are different in these compartments. Further, once the defective viruses are formed, they do not propagate. This is reminiscent of endogenous retroviruses in the human genome that reside as multiple defective viral sequences

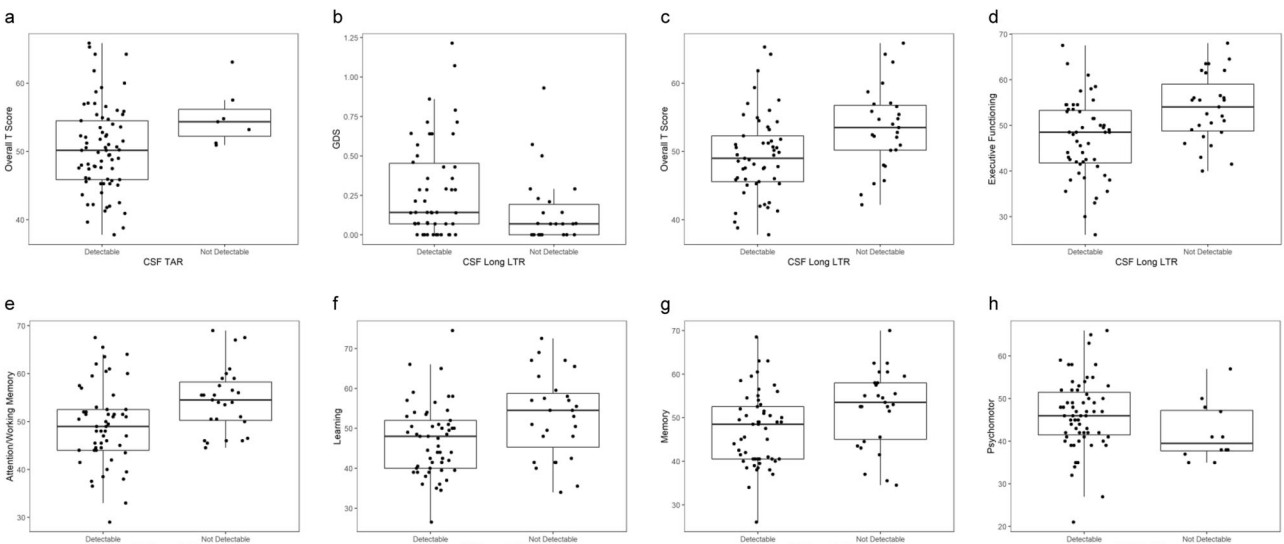

**Fig. 2 | Detectability of EV-associated HIV RNA is associated with neurocognitive deficits.** The ddPCR assay measurements (n = 84) were binarized into detectable vs. non-detectable against clinical outcome and significance was tested using a two-sided Wilcoxon rank sum test for each compartment with no adjustments for multiple comparisons. The significant relationships are depicted in (**a**–**h**). **a** TAR vs. T-score p = 0.034, **b** Long LTR vs. GDS p = 0.018, **c** Long LTR vs. T-score p = 0.004, **d** Long LTR vs. Executive Functioning p = 0.001, **e** Long LTR vs. Attention and Working Memory p = 0.006, **f** Long LTR vs. Learning p = 0.011, **g** Long LTR vs. Memory p = 0.011, and **h** Tat-Rev vs. Psychomotor p = 0.018. The center represents the median, boxes represent 1st to 3rd quartile, and the whiskers represent the upper and lower fences (±1.5*IQR). CSF cerebrospinal fluid, LTR long terminal repeat. Source data are provided as a Source Data file.

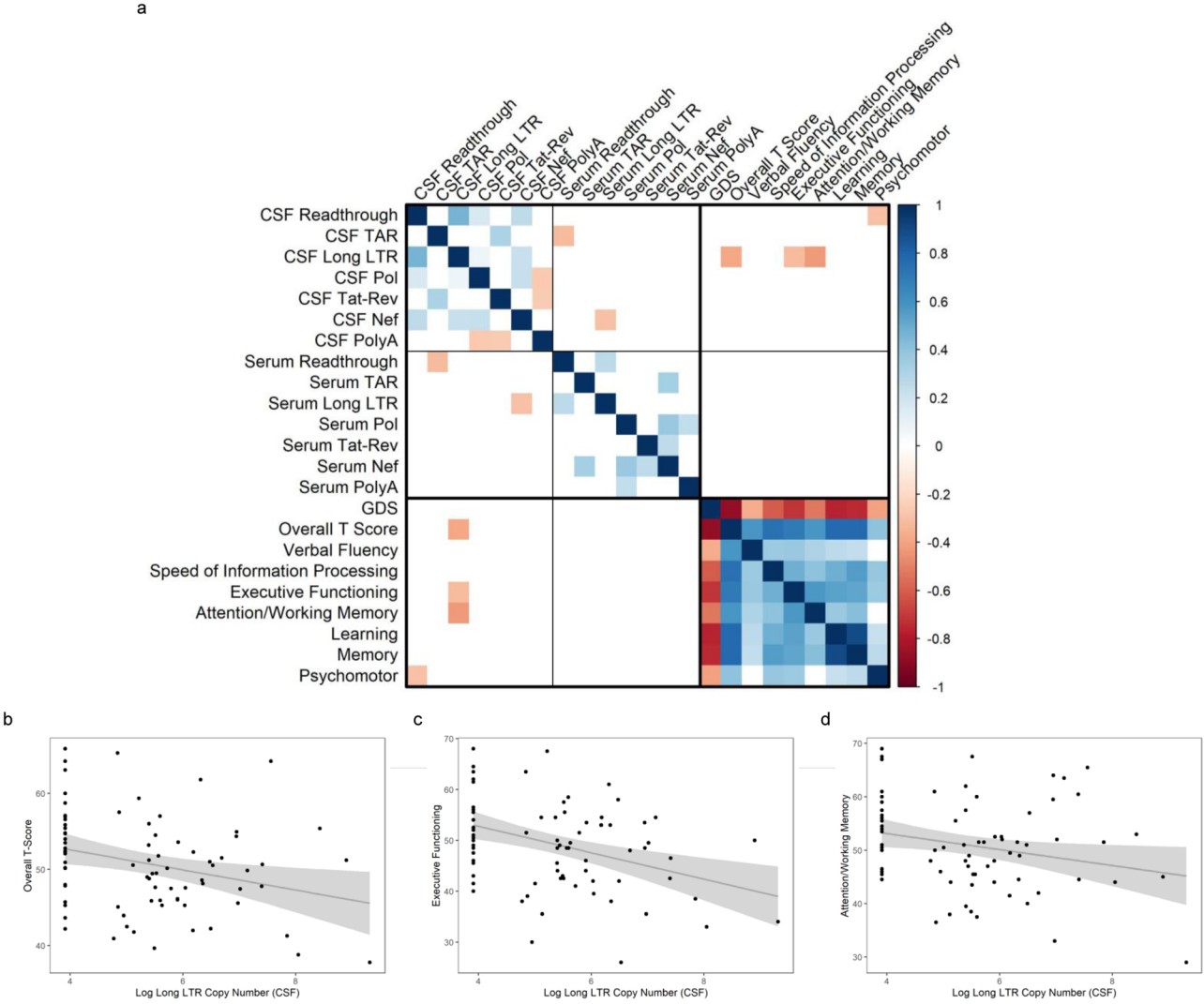

**Fig. 3 | EV HIV RNA copy number is correlated with neurocognitive impairment. a** Spearman correlation matrix was used to visualize the relationships specifically between RNA copy numbers and clinical outcomes ($n = 84$). The correlation matrix uses color (blue = positive; red = negative) to depict the two-sided, pairwise Spearman correlation results (without adjustments for multiple comparisons). Significant two-sided correlations were shown, while non-significant correlations were omitted (white). Pairwise tests that were significant at the 0.05 level were further investigated graphically and in a linear model predicting clinical outcomes, first unadjusted, and then adjusted for covariates (sex, age, duration of ARV treatment, duration of HIV infection, nadir CD4, and viral load). CSF Long LTR was significantly associated with (**b**) overall T-score ($r = -0.28$, $p = 0.010$), (**c**) Executive Function ($r = -0.36$, $p = 0.001$), and (**d**) Attention and Working Memory ($r = -0.22$, $p = 0.047$). The gray region represents pointwise 95% confidence intervals and gray line represents the unadjusted $r$. CSF cerebrospinal fluid, TAR transactivation response element, LTR long terminal repeat, GDS global deficit score. Source data are provided as a Source Data file.

some of which have open reading frames but none of which can form a replicating viral particle or propagate[26]. Conversely, full length viral sequences were concordant between serum and CSF suggesting that the two compartments can seed one another.

Interestingly, there appears to be a gradient for detectability of the viral transcripts in the CSF, such that the ones closer to the transcription start site were more likely to be detected suggesting they were more abundant. This could be due to non-processive transcription due to elongation blocks which affect the efficiency at multiple steps that are a result of a myriad of mechanisms involving (1) host transcription factors, (2) chromatin remodelers, modifiers and chaperones, (3) host negative regulators of transcription such as NELF, and (4) histone positioning[18,27–29]. Alternatively, or perhaps concurrently, the higher abundance of transcripts with closer proximity to the transcription start site could be due to a high proportion of 3' defective proviruses within the CNS reservoir. This is in line with previous findings from viral sequences in the periphery that suggest that the regions closer to the 5' LTR are more conserved than those located

from pol onward[30]. This is potentially due to selective pressures toward the 3' end of the genome to evade reverse transcriptase, integrase, and protease inhibitors, which lead to underdetection of the transcript in this assay. However, there is evidence to suggest that as the proviral landscape shifts during the course of infection there is a lower percentage of hypermutated proviruses and a higher percentage of proviruses with large internal deletions[31]. Further study is needed to fully characterize the proviral sequences of the CNS and re-evaluate the distribution of defective regions with the introduction of capsid inhibitors such as Lenacapavir into ART regimens.

The Long LTR amplicon correlated with changes in neurocognitive functioning both in the cross-section and longitudinal cohorts. The primers for this assay amplify a highly conserved region of the HIV LTR (HXB2 position 522-643) which is predicted to have minimal deletions and/or mutations and is thought to measure nearly 100% of all proviruses (intact and defective)[30]. The correlation of an RNA amplicon, transcribed from both intact and defective proviruses, with neurocognitive functioning suggests that both intact and defective

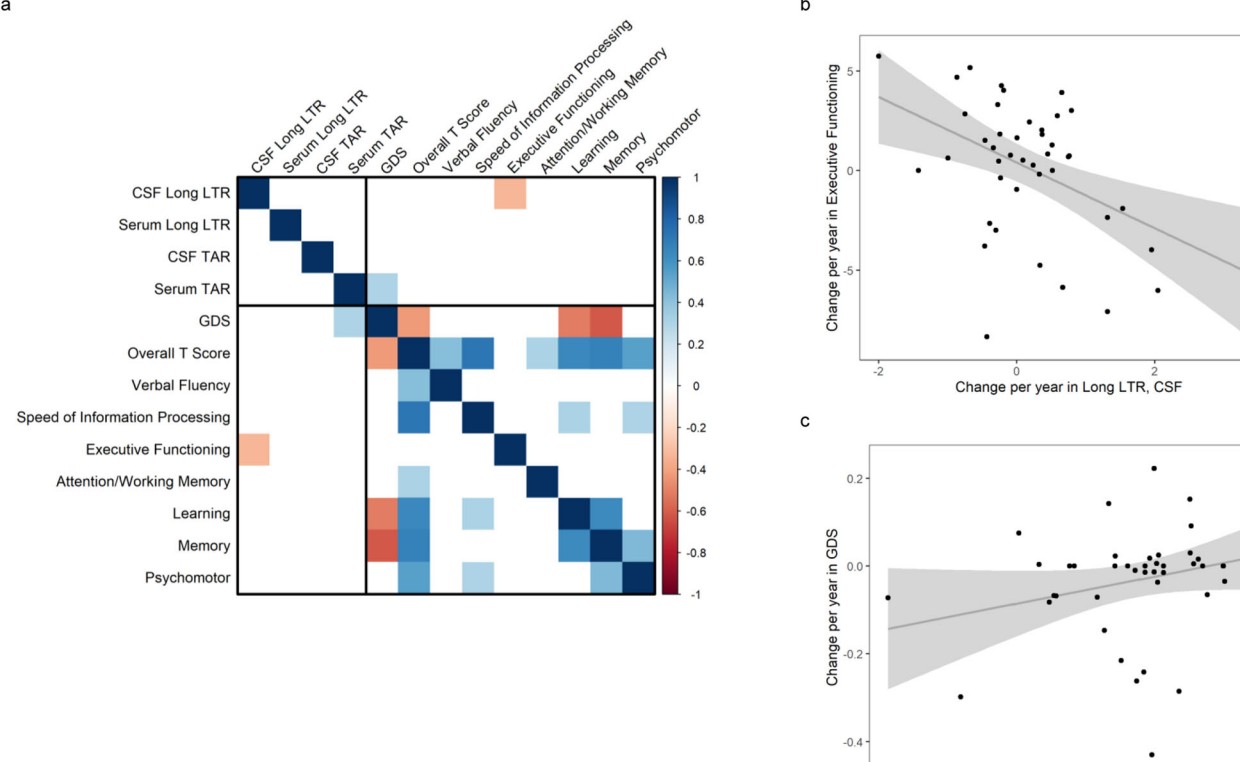

**Fig. 4 | Changes in EV-associated HIV RNA is correlated with changes in neurocognitive dysfunction. a** Correlation matrix using Spearman's correlations between linear model-estimated slopes (change per year) for each individual ($n = 43$) on each parameter across time. The correlation matrix uses color (blue = positive; red = negative) to depict the two-sided pairwise Spearman correlation (without adjustments for multiple comparisons) results. Significant correlations were shown, while non-significant correlations were omitted (white). **b** Negative correlation between the change in CSF Long LTR copy number and the change Executive Functioning domain score over time ($r = -0.332$, $p = 0.03$). **c** Positive correlation between the change in Serum TAR copy number and the Global Deficit Score over time ($r = 0.309$, $p = 0.044$). The gray region represents pointwise 95% confidence intervals and gray line represents the unadjusted $r$. CSF cerebrospinal fluid, TAR transactivation response element, LTR long terminal repeat, GDS global deficit score. Source data are provided as a Source Data file.

proviruses may contribute to neuropathophysiology in HIV infection. This finding is critical, as recent reports have suggested that less than 10% of the ~1000 proviruses/10[6] cells found in the frontal cortex of PLWH were intact[2,4]. However, despite making up over 90% of the proviruses in the brain, defective proviruses can have open reading frames and produce neurotoxic viral proteins including Env, Tat and Nef[8]. These viral proteins can be incorporated into EVs alongside the viral RNAs and can contribute to neurotoxicity, neuroinflammation as well as damage to the blood brain barrier[8,32,33]. Interestingly, levels of Long LTR most reliably correlated with decreases in performance on tests which measure executive functioning and was also reflected in the composite scores of neurocognitive dysfunction (Global Deficit Score and T-Score). Previous studies have also shown impairment of executive functioning in PLWH treated with ART[34–37]. This may suggest increased vulnerability of these regions in the prefrontal cortex to viral products in this population.

Although the Long LTR is predicted to be primarily intact, the cloned sequences from the CSF and serum showed that most individuals had one intact sequence which was found in both the CSF and serum and deleted/mutated sequences that were unique to each compartment, suggesting that the seeding of the CNS reservoir from the periphery and subsequent mutation within the compartment with no crosstalk. Conversely, in two individuals two sequences were found to be shared between the CSF and serum, one intact and one deleted/mutated. These findings may indicate reseeding of the periphery by the CNS compartment. Future long-read sequencing studies are needed to confirm these observations. Interestingly, the Long LTR deletions resulted in the loss of the primer activation site, the tRNA

stabilizing motif, and the anticodon binding motif and excluded the polyA site and PBS. This deletion likely results in a drastic decrease in reverse transcription efficiency as these sites primarily have a stabilizing function, but may not abrogate it altogether[38,39], allowing for low level virus replication and transcription that potentially goes undetected by the immune system. However, our sequencing approach likely selected for transcripts with intact PBS and polyA sites and therefore the presence of polyA and PBS deletions cannot be ruled out without further study. Sequencing of the HIV *env* C2-V3-C3 region showed a similar pattern to that of the Long LTR with a predominately intact sequence common to the CSF and serum. Interestingly, a number of serum sequences were deleted in the V3-C3 region, which contains epitopes critical for cytotoxic T-lymphocyte activity, N-linked glycosylation, tropism, and CD4 binding suggesting extensive immune evasion in the periphery. These same deletion mutants were not found in the CSF, suggesting the possibility of compartmentalization of the viral reservoirs.

Remarkably, the multiply spliced Tat-Rev transcript was found in a high copy number relative to other transcript levels in the CSF, which is contrast with the reported transcript levels in CD4+ T-cells from HIV patients in which Tat-Rev was found to be the lowest in abundance[18]. In that study, the patients were not fully controlled on ART, hence it is possible that complete control of viral replication by ART may result in increased spliced transcripts as shown previously[40]. Another possibility is that multiply spliced transcripts are actively transported out of the cell through a yet-to-be described mechanism. It may also suggest that the CNS has a relatively high number of 5' defective proviruses, which would be in line with recent reports which showed that defects

in the major splicing donor (MSD) site caused a reduced efficiency in RNA dimerization and nucleocapsid binding allowing for a shift toward a higher relative abundance of multiply spliced transcripts[41]. This and the correlation of the multiply spliced transcripts with changes in the psychomotor domain require further study.

Viral RNA was detected in all samples tested despite adequate antiretroviral therapy and an undetectable viral load. This is in contrast to other large-scale studies which estimate RNA to be detectable in ~42% of CSF and 65% of plasma[42]. This discrepancy is likely due to the front-end enrichment of EVs from CSF and serum which allows for a more sensitive assay as compared to nucleic acids isolated directly from the CSF or serum. Additionally, this study utilizes 7 different ddPCR assays which span the HIV genome and allows for detection of defective viral sequences. Even though we do not know what percentage of viral transcripts from infected cells in the brain get incorporated into the EVs, in vitro studies suggest that the viral RNA profile in EVs corresponds to the transcript levels within the originating cells in myeloid lineage cells in the presence and absence of antiretroviral therapy[9]. Nonetheless, our findings suggest that HIV transcription in viral reservoirs like the brain is likely underestimated by the current literature. While the use of a bulk isolation method was effective in increasing detectability, a limitation of this study is the use of bulk CSF and serum EVs as a marker for transcriptional activity in the brain and peripheral blood reservoirs without distinguishing between cell type reservoirs. Recent work has shown that resident microglia in the brains of HIV-infected individuals contain replication-competent proviruses and serve as a persistent viral reservoir in the presence of antiretroviral therapy[43]. However, lymphocytes and monocytes can traffic into the CNS and therefore could contribute to the production of viral RNAs detected in the CSF alongside astrocytes which have restricted viral replication. EVs also have the potential to traverse the blood brain barrier, and therefore, transcripts arising from other latent reservoirs cannot be ruled out. To date, there is no gold standard for the isolation of cell type-specific EVs from complex samples and further work is needed in the field to refine EV isolation to determine the cell type of origin. Additional studies should include comparisons of EV viral RNA from specific cell types paired with single cell RNA sequencing of postmortem brain tissue.

These findings have important implications for treatment strategies. To achieve a cure, all viral transcription needs to be blocked. Currently available ARTs fail to impact the production of viral transcripts from the viral reservoirs, despite high CNS penetrance of some antiretroviral drugs. Our data also suggests that strategies to block HIV transcription should be targeted to the regions involved in initiation of transcription. These include gene excision, chromatin remodeling and use of RNA interference strategies.

## Methods

All research complies with all relevant ethical regulations and has been approved by the NIH Institutional Review Board (IRB). Informed consent was obtained from all individuals. Financial compensation is provided to participants as per NIH IRB recommendations.

### Patient population

The ALLHANDS study (ClinicalTrials.gov ID: NCT01875588; NIH Clinical Center, Bethesda, MD, protocol 13N0149) is a 20-year natural history study of 300 HIV+ individuals, as well as 150 HIV− controls. Each patient is followed annually for a total of 10 years. Visits include history and physical, blood and urine collection, neuropsychological testing, a clinical depression questionnaire, MRI scans of the brain and an optional lumbar puncture and/or ophthalmology exam. Common comorbidities in the tested cohort ($n = 84$) include history of depression (30%), hypertension (45%), hyperlipidemia (29%), anxiety (22%), type II diabetes (19%), and chronic hepatitis C (15%). One individual had a remote small basal ganglia infarct.

Inclusion criteria for HIV+ patients were (1) documented HIV-1 infection, (2) plasma HIV-RNA < 50 copies/mm³ or below level of detection for greater than 1 year, and (3) at least 1 year of continuous ART. Exclusion criteria included (1) illness or other condition that, in the opinion of the principal investigator, may interfere with study participation at the time of enrollment, including, but not limited to CNS infections and non-CNS opportunistic infections; (2) conditions other than HAND associated with cognitive impairment or dementia such as Alzheimer's, Parkinson's disease, head injury with loss of consciousness >30 min, untreated sleep apnea with day-time sleepiness, or seizure disorders; (3) concurrent severe, unstable psychiatric illness that, in the opinion of the investigators, may interfere with study participation and/or data interpretation; (4) concurrent substance abuse that, in the opinion of the investigators, may interfere with study participation and/or data interpretation; (5) contraindications to MRI scanning; (6) use of narcotics, psychiatric, and anti-seizure medications; (7) inability to refrain from use of anticoagulant/antiplatelet medication; (8) prior or planned/anticipated exposure to radiation due to clinical care or participation in other research protocols; and (9) pregnant or lactating females.

### Neuropsychological testing

Participants underwent an annual battery of neuropsychological tests administered by a trained psychometrist and overseen by a licensed and board-certified neuropsychologist. Fifteen primary measures of cognition, subsumed under seven domains, including speed of information processing (SIP), working memory, learning, memory, executive, verbal, and motor functioning were administered. Raw scores derived from those measures were converted to T-scores while calibrating for age, education, gender, and/or race. T-scores were combined to create seven domain T-scores. An overall T-score was generated from the average score of all fifteen measures. A Global Deficit Score (GDS) was also compiled as a summation of the presence and severity of each participant's deficit scores.

### Extracellular vesicle enrichment and RNA isolation

Extracellular vesicles were enriched from 500 μl neat CSF/serum using NanoTrap particles (Ceres Nanosciences) as previously described[8,9]. Briefly, 30 μl of Nanotrap particles was added to 500 μl neat CSF/serum and rotated overnight at 4 °C. Samples were centrifuged at $20,000 \times g$ at 4 °C and depleted CSF/serum was discarded. EV RNA was then isolated using Trizol according to manufacturer's protocol (Invitrogen), and the RNA was treated with Turbo DNase (Invitrogen) as per manufacturer's instructions to eliminate any remaining DNA. RNA concentration was quantified using ultraviolet (UV) spectrophotometry (NanoDrop 1000). This method leads to bulk isolation of EVs and does not provide information on the distribution of the different types of EVs. We chose this method to reduce manipulation of the sample, maximize detectability of the viral RNAs present in each compartment, and provide an overview of the viral RNA profile within the compartment as a surrogate measure of transcription within that reservoir. The individuals included in this study were well controlled on antiretrovirals and have undetectable viral loads. Therefore, EV preparations are unlikely to include viral particles.

### Polyadenylation-reverse transcription

To increase the detection efficiency of short RNA transcripts containing the TAR region (a tight hairpin loop structure), a separate aliquot of RNA (comprising about 20% of the total RNA) was polyadenylated as previously described[18]. Briefly, 1 μg of RNA was added to 3 μl of 10X RT buffer (Invitrogen), 3 μl of 50 mM $MgCl_2$ (BioRad), 1 μl of 10 mM adenosine 5′-triphosphate (Lucigen), 2 μl of polyA polymerase (4 U/μl, Lucigen), and 1 μl of RNAseOUT (40 U/μl, Invitrogen) and incubated at 37 °C for 45 min. The remaining reagents for the RT reaction were then

added, including 1.5 µl of 10 mM dNTPs (Invitrogen), 1.5 µl of random hexamers (50 ng/µl, Invitrogen), 1.5 µl of 50 µM oligo dT15, 1 µl of SuperScript III reverse transcriptase (200 U/µl, Invitrogen), and 4.5 µl of PCR-grade water for a total of 10 µl. The reaction was incubated in a thermocycler at 25 °C for 10 min, 50 °C for 50 min, followed by an inactivation at 85 °C for 5 min.

## Reverse transcription

For all other ddPCR assays except the TAR assay, a 1 µg aliquot of RNA was used in a conventional RT reaction. The reaction was performed in a 50 µl mixture containing 5 µl of 10X RT buffer (Invitrogen), 5 µl of 50 mM MgCl2 (BioRad), 2.5 µl of 50 µM oligo dT15, 2.5 µl of 10 mM dNTPs (Invitrogen), 2.5 µl of random hexamers (50 ng/µl, Invitrogen), 1.25 µl of RNAseOUT (40 U/µl, Invitrogen), 2.5 µl of SuperScript III reverse transcriptase (200 U/µl, Invitrogen), and 28.75 µl of PCR-grade water. To allow for measurement of all 7 sequence regions, all reagents were increased proportionally to a total volume of 70 µl. "No RT" negative controls were established via the above method from patient RNA with the exception of the addition of the SuperScript III reverse transcriptase. Positive controls were made with RNA extracted from HIV-1 LAV infected Jurkat E6 Cells (J1.1).

## Droplet digital PCR

Primer and probe sequences were previously developed by Yukl et al.[18] (Supplementary Data Table 4). Digital droplet PCR was performed using the QX100 Droplet Digital qPCR System (Bio-Rad) with each sample and control tested in triplicate. Each sample well contained 20 µl of a mix composed of 12.5 µl of ddPCR Probe Supermix-no dUTP (Biorad), 0.45 µl of forward primer at 50 µM, 0.45 µl of reverse primer at 50 µM, 0.625 µl of probe at 10 µM, and 5.975 µl of PCR-grade water. Five microliters of cDNA was added to each sample well for a total of 25 µl per well. The No RT negative controls and the J1.1 (HIV-1 lymphadenopathy-associated virus (LAV)-Infected Jurkat E6 cells (J1.1), ARP-1340 obtained through the NIH HIV Reagent Program, Division of AIDS, NIAID, NIH, contributed by Dr. Thomas Folks) positive controls were included on each plate.

After the droplets were generated in each well, they were amplified with a thermocycler for 10 min at 95 °C, 45 cycles of 30 s at 95 °C and 59 °C for 60 s, and a final step of 10 min at 98 °C. Droplets were read and analyzed using the QuantaSoft software in the absolute quantification mode. The ddPCR assays utilized have been extensively validated and characterized previously[18]. However, the detection limit was validated using low passage 8e5 cells (AIDS Reagents Program, ARP-95) to create a series of DNA standards for assay validation.

## Sequencing

Isolated EV RNA from matched CSF and serum samples was reverse transcribed as described above. The cDNA was diluted 1:4 with water and amplified with Q5 High-Fidelity Master Mix (New England BioLabs M0492S) according to manufacturer's protocols (see Supplementary Data Table 4 for primers). Samples were run on a 1.2% agarose gel at 80 V for 40 min. The presence of a single band was confirmed, and concentration was quantified using ultraviolet spectrophotometry (NanoDrop1000). PCR product with 3′-A overhangs was generated with Premix Ex Taq (Takara Bio, RR003A) according to manufacturer's protocol. PCR product was cloned into the pCR4-TOPO TA vector using the TOPO TA Cloning kit (Thermo Fisher, K457501) according to manufacturer's protocol. One Shot Top10 Chemically Competent *E. coli* were transformed and grown overnight at 37 °C. At least 10 colonies per sample were selected, grown overnight at 37 °C, and purified using the QIAprep Spin Miniprep Kit (Qiagen, 27104). Sanger sequencing was used to obtain amplicon sequences. DNA and amino acid sequences were aligned and visualized using Clustal Omega in DNAStar Lasergene MegAlign Pro 17.

## Statistical analysis

For cross-sectional analyses, detectable viral RNAs for each compartment were compared with a Student's *t* test (Fig. 1). A detectability analysis was also performed, binarizing assay measurements into detectable vs non-detectable levels against clinical outcome and testing with Wilcoxon Rank-Sum test (Fig. 2). A Spearman correlation matrix was used to visualize the relationships specifically between assay measurements and clinical outcomes. The correlation matrix uses color (blue = positive; red = negative) to depict the pairwise Spearman correlation results. Significant correlations were shown, while non-significant correlations were omitted (white), for simplicity of visualization. Pairwise tests that were significant at the 0.05 level were further investigated graphically and in a linear model predicting clinical outcomes, first unadjusted, and then adjusted for covariates (sex, age, duration of ARV treatment, duration of HIV infection, nadir CD4, and viral load). Assay copy number was the key independent variable and was log-transformed in the model after adding 50 units to remove log-transformation of a zero score (Fig. 3).

For longitudinal analyses, a similar correlation matrix was generated, using correlations between linear model-estimated slopes (change per year) for each individual on each parameter across time. Assay measurements were log-transformed. Fully complete data was used to ensure matching/consistency of timepoints within individual for the most cohesive data. Significant correlations were further investigated graphically (Fig. 4).

A Pearson correlation was used to investigate the relationship between EV quantities, viral RNA copy numbers, and neuropsychological data (Supplementary Data Fig. 1).

Analyses were conducted in R version 4.2.0. and GraphPad Prism version 9.

## Reporting summary

Further information on research design is available in the Nature Portfolio Reporting Summary linked to this article.

## Data availability

The sequence data generated in this study have been deposited Gen-Bank with the accession numbers as follows: Long LTR (PP760520-PP760729) and Envelope sequences (PP760730-760985). The HIV HXB2 reference genome accession number is K03455 M38432. Source data are provided with this paper.

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

## Acknowledgements

This research was supported by the Division of Intramural Research of the NIH, NINDS (NS3130). The content is solely the responsibility of the author(s) and does not necessarily represent the official views of the National Institutes of Health.

## Author contributions

C.D. and A.N. designed the study. C.D., J.D., M.C., and L.H. designed and conducted experiments. D.D., D.P., B.S., and A.N. did the clinical char-acterization. G.N. provided statistical analysis and support. E.G. and J.S. conducted the extensive neuropsychological testing. C.D., J.D., L.H., and A.N. analyzed data and wrote the manuscript. All authors provided intellectual content.

## Funding

## Competing interests
The authors declare no competing interests.
