## [Peer Review File · Nature Communications]

HIV-1 RNA in Extracellular Vesicles is Associated with Neurocognitive OutcomesReviewers' Comments:

Reviewer #1:

Remarks to the Author:

The manuscript entitled "HIV-1 RNA in Extracellular Vesicles is Associated with Neurocognitive Outcomes" demonstrates the presence of HIV RNAs in the EVs isolated from matched CSF and serum samples of people living with HIV (PLWH). Seven digital droplet polymerase chain reaction assays (ddPCR) were used to assess the transcriptional status of the latent reservoir. The findings demonstrated that there was increased abundance of the EV-associated vRNA in the CSF and that cross-sectionally and longitudinally correlated with neurocognitive dysfunction in a cross sectional study of 84 HIV+ individuals and in 43 followed longitudinally. This is a very important study that underscores the role of persistence virus transcription in virologically suppressed PLWH, thereby contributing to the various phenotypes of HAND and sheds light on the differential selective pressures on the virus in the two compartments. Overall a highly significant, well-written and a timely study that adds vital information on the regions critical for initiation of viral transcription that could be blocked for treatment strategies. There are some minor concerns that need to be addressed.

1. Can the authors mention if there were any other associated comorbidities in the tested cohort.
2. Please clarify and discuss if any gender specific differences were observed in the HIV-1 RNA copy numbers in EVs isolated from CSF/plasma samples of male vs. female PLWH.
3. Authors have correlated the presence of HIV-1 RNA transcripts in the EVs from CSF, and plasma samples to the long-term HAND implications. Can the authors comment on the role of different viral proteins that could also be present in these EVs.
4. Can the authors provide what percentage of the viral RNA is present in the EVs. Is it significant enough in comparison to the viral transcripts found in the brain? This is a comment and not a concern.
5. While microglia are the key virus reservoirs in the brain, is it possible that astrocytes could also be a source of these vRNAs. This should be discernable. Again, it is a comment, not a concern.

Reviewer #2:

Remarks to the Author:

Overall, this manuscript describes cross-sectional and longitudinal studies of EV RNA content potential correlation with neurocognitive disorders in PLWHs.

In comparing the content RNA EV cargo of matched plasma and CSF from PLWHs, the authors have detected HIV mRNA transcripts within EVs of virally suppressed PLWHs. These results are interesting and noteworthy because they are the first to report that specific HIV transcripts can be detected in EVs derived from sera or CSF of virally suppressed PLWHs in different proportions. This work could have a significant impact on the fields of EV biology and HIV neuropathology. The work further supports previous studies describing EV content from PLWHs. Other studies on PLWH EV content have found HIV TAR within EVs (Kashchani et al 2014?). So the concept of HIV-related RNAs within EVs is well-founded.

The methodology using nano traps is sound and has been used by several research groups in the EV field.

However, the isolation and characterization of the EVs should be better described. The authors do not define the type of EV is detected in the serum and CSF – exosomes vs. microvesicles. It would be interesting to see whether the distribution of EV types differs in serum and CSF in the context of HIV infection. Also, the investigators should show or mention whether the EVs were characterized and provide evidence of the EV subsets' isolation. This can be done via western blot, TEM, flow cytometry, etc.

There may be a typo: Line 51 – please define eCNS. (typo?)

Given that HIV and exosomes are similar in size, The authors should perform tests that discriminate virus from EV. Besides minor details, this work is very interesting and comparable to the EV field work. Assuming the authors have only isolated EVs, the work is noteworthy, and the data analysis, interpretations, and conclusions seem correct. Viral RNA within EVs represents an exciting new

prospect for how lentiviruses, such as HIV, disseminate and induce pathology.

Reviewer #3:

Remarks to the Author:

DeMarino et al. used ddPCR targeting different portions and forms of HIV transcripts in serum and CSF samples from a mixed cross-sectional/longitudinal cohort of people living with HIV to describe the link between neurocognitive dysfunction and HIV transcripts detection/levels in blood and CSF extracellular vesicles. The topic is of interest as the burden of HIV-associated neurocognitive disorders even when antiretroviral is successful remains a challenge in HIV infection care. The methodology is sound. The results presented here are relevant in the field, the manuscript is well written. However, I think some information and analyses are missing, but can be further added, and the conclusions should sometimes be re-drawn, in regards to some points I mentioned below. Thus, I highlight here some major and minor points that I would like the authors to address.

Major Comments

A. My first point here is a reflection about what extracellular vesicles mirrors regarding the HIV reservoir. In other terms, do HIV transcripts detected in exosomes reflect the transcriptional activity of the viral reservoir? Would there be a possibility for a cell to transcribe HIV but not produce exosomes (or to to lesser a more important extent than compared to another cell)? Besides, extracellular vesicles (EV) can cross the blood brain barrier (BBB). So there might be a possibility that HIV transcripts detected in EV do not arise from the compartment in which they are detected. This could be shortly discussed to dampen some conclusions.

B. Patient characteristics: It was rather difficult for me to understand if participants were treated (and the current drugs) or not and if they were in virological success. I could find some treatment information for the 43 longitudinal participants in extended table 1. But it's missing for the remaining 41 participants. I think this information should be added in Table 1. Same for the drugs, I think the numbers aren't quite right, it is stated that 6 participants are "currently on an integrase inhibitor", however when I recounted the number of participants treated with an integrase inhibitor in the extended table 1 for the 43 participants, at least 31 of them are currently on an integrase inhibitor. Please check all the numbers presented in this table. Besides the NNRTIs class and some NRTI drugs are missing in Table 1. Finally, viral load information is missing as well: are some of these participants viremic? This is really an important point that should be added for better understanding and interpreting the results presented here.

C. I think the author present very nice data in Fig1C-I, for which more analyses and conclusions could be drawn. First, a minor point for clarity: The authors mention that transcripts were not detected in all samples in Fig 1J but this does not appear in Fig1C-I. Are they plotted? If not please do so and place them at the limit of detection. If yes maybe plot them with an open circle or another symbol? Second, even though correlation between both compartments in paired samples and for each transcript is described in the extended data. It would be nice to say whether it is the case or not in the results section. Third, the difference or absence of difference in HIV transcripts levels varies according to the transcript analyzed. Would the authors think that transcriptional blocks are different between blood and the CNS? I think it should be nice to assess that by performing paired comparisons between transcripts (to be performed separately for CSF and serum) like in the original Yukl paper. The authors mention these transcriptional blocks in the discussion section, but they did not perform statistical tests to prove it, but actually this seems quite feasible and will surely lead to some nice conclusions. Fourth, Tat-Rev transcripts were observed to be at the lowest levels in the Yukl paper, here it seems at least in the CSF that Tat-Rev levels are as high as pol levels (in the CSF as well). I think it would be worth mentioning especially that the authors found an inverse correlation with the Psychomotor domain. This is quite interesting, and it would worth digging in the literature the implication of early and late HIV transcripts.

D. Some CSF HIV transcripts levels correlate with neurocognitive disorders scores. Do the authors have a hypothesis on why this is only the case for readthrough and long LTR? They mention the

conservation of the LTR target sequence across HIV clades in the discussion section, so does this mean that for other transcripts with a more variable sequence there could have been an underdetection thus leading to an absence of correlation with neurocognitive disorders scores?

E. Compartmentalization is hard to prove especially with few samples and few sequences retrieved. Besides, can we talk about compartmentalization in case EV containing HIV transcripts. The reservoir might be distributed similarly between CNS and blood but its transcriptional profile might be different. I would soften conclusions here. About the same intact sequences were found between serum and CSF, and the authors hypothesized seeding between those compartments. Could there be a passive transfer or is it local replication highly similar variants?

Minor Comments

1. Viral particles present in the blood (measured routinely for viral load assessments) were shown to arise mostly from the lymph node HIV-infected cells, would the authors think the serum EVs might come from there too? Is there a way to assess that maybe by phenotypic analysis of EVs?
2. Do the authors have an idea on HIV clade distribution across participants? In other terms might some given clades be more frequently associated with HIV transcripts detection in CSF and/or neurocognitive disorders?
3. Fig3B-D Analyses were adjusted to viral load but what is the viral load (see my point above) are these individuals not supposed to be controlled? If so, why adjust to viral load? Is some of these individuals are viremic it would seriously alter the conclusions that were made.
4. Serum readthrough seem to inversely correlate with CSF TAR, same for serum long LTR and CSF nef. Is there an hypothesis behind this?
5. Some antiretroviral drugs successfully pass the BBB? Might drug levels in serum/CSF have a relation with HIV transcripts levels? If ART could influence transcription and in this case residual replication so maybe adjust to ART (as some drugs would better pass through the BBB than others) as well?
6. I think it would be nice for the readers to better plot the longitudinal data: maybe the time since undetectability in x axis and the HIV transcript levels in y axis and link points from a given participant to see individual trajectories. Thus we could see what are the trends: some patient increase their levels, some not.

REVIEWER COMMENTS

Reviewer #1 (Remarks to the Author):

The manuscript entitled “HIV-1 RNA in Extracellular Vesicles is Associated with Neurocognitive Outcomes” demonstrates the presence of HIV RNAs in the EVs isolated from matched CSF and serum samples of people living with HIV (PLWH). Seven digital droplet polymerase chain reaction assays (ddPCR) were used to assess the transcriptional status of the latent reservoir. The findings demonstrated that there was increased abundance of the EV-associated vRNA in the CSF and that cross-sectionally and longitudinally correlated with neurocognitive dysfunction in a cross sectional study of 84 HIV+ individuals and in 43 followed longitudinally. This is a very important study that underscores the role of persistence virus transcription in virologically suppressed PLWH, thereby contributing to the various phenotypes of HAND and sheds light on the differential selective pressures on the virus in the two compartments. Overall a highly significant, well-written and a timely study that adds vital information on the regions critical for initiation of viral transcription that could be blocked for treatment strategies. There are some minor concerns that need to be addressed.

1. Can the authors mention if there were any other associated comorbidities in the tested cohort.

We have added a description of the comorbidities in the tested cohort on lines 318-320. We have also included additional analyses (Extended Fig. 2) which examined differences in viral transcripts between comorbidity groups. This has been included in the text on lines 102-109.

2. Please clarify and discuss if any gender specific differences were observed in the HIV-1 RNA copy numbers in EVs isolated from CSF/plasma samples of male vs. female PLWH.

To address this comment, we have performed Wilcoxon Rank Sum testing on each of the viral transcripts (7 CSF and 7 serum) to determine any differences in viral RNA copy numbers between males and females. This has been included in the results section of the manuscript on lines 124-127, “Each transcript was further investigated using Wilcoxon Rank Sum test to investigate differences in viral transcripts between males and females. None of the transcripts differed significantly between males and females”.

3. Authors have correlated the presence of HIV-1 RNA transcripts in the EVs from CSF, and plasma samples to the long-term HAND implications. Can the authors comment on the role of different viral proteins that could also be present in these EVs.

Yes, the persistence of viral transcription despite antiretroviral therapy contributes to the presence of viral proteins in various reservoirs during adequate viral suppression. These proteins have been found in extracellular vesicles and have been implicated in some of the chronic pathogenic mechanisms observed in PLWH. We agree that addition of this information would nicely add to the discussion of the results. We have included this on lines 223-227.

4. Can the authors provide what percentage of the viral RNA is present in the EVs. Is it significant enough in comparison to the viral transcripts found in the brain? This is a comment and not a concern.

The reviewer raises a great question that we believe would best be answered using single cell sequencing of postmortem brain tissue paired with long read sequencing of perimortem CSF and blood. A study to address this comment would need to be carefully designed to ensure adherence to antiretrovirals through time of death, short post-mortem intervals to yield high quality RNA in the brain tissue, and collection of CSF and serum perimortem but prior to tissue collection. We have added a sentence to the discussion (lines 286-289).

5. While microglia are the key virus reservoirs in the brain, is it possible that astrocytes could also be a source of these vRNAs. This should be discernable. Again, it is a comment, not a concern.

Yes, we do believe that astrocytes could contribute to the production of viral RNAs that we have detected. We have included this in our discussion on lines 272-284. We hope that as the field progresses cell type origin isolation becomes a feasible approach to help answer some of these major questions in the field.

Reviewer #2 (Remarks to the Author):

Overall, this manuscript describes cross-sectional and longitudinal studies of EV RNA content potential correlation with neurocognitive disorders in PLWHs.

In comparing the content RNA EV cargo of matched plasma and CSF from PLWHs, the authors have detected HIV mRNA transcripts within EVs of virally suppressed PLWHs. These results are interesting and noteworthy because they are the first to report that specific HIV transcripts can be detected in EVs derived from sera or CSF of virally suppressed PLWHs in different proportions. This work could have a significant impact on the fields of EV biology and HIV neuropathology. The work further supports previous studies describing EV content from PLWHs. Other studies on PLWH EV content have found HIV TAR within EVs (Kashchani et al 2014?). So the concept of HIV-related RNAs within EVs is well-founded. The methodology using nano traps is sound and has been used by several research groups in the EV field.

However, the isolation and characterization of the EVs should be better described. The authors do not define the type of EV is detected in the serum and CSF – exosomes vs. microvesicles. It would be interesting to see whether the distribution of EV types differs in serum and CSF in the context of HIV infection. Also, the investigators should show or mention whether the EVs were characterized and provide evidence of the EV subsets' isolation. This can be done via western blot, TEM, flow cytometry, etc.

We agree with the reviewer that it would be interesting to see whether the distribution of EV subtypes differs between the serum and CSF particularly in the context of HIV infection and viral RNA profiles. We believe that this could be valuable information to the field of EV biology. However, for the current study we have chosen a bulk isolation method (i.e. Nanotraps) to A) reduce manipulation of sample to maximize detectability of the viral RNAs present in each compartment and B) provide an

overview of the viral RNA profile within the compartment as a surrogate measure of transcription within that reservoir. Given the complexities that exist in the incorporation of various cargos into different subtypes of extracellular vesicles we believe that subset isolation would provide skewed data and not represent an unbiased snapshot of viral transcription within the reservoir. As the reviewer mentioned, the methodology using Nanotrap is sound and has been used and validated by several groups in the EV field and specifically by groups investigating the role of EVs in retroviral infections (Jaworski et al. 2014, Sampey et al. 2016, Barclay et al 2017, DeMarino et al. 2018, Anderson et al. 2018, Barclay et al. 2018, Pinto et al. 2019, DeMarino et al. 2019, Henderson et al. 2019, Branscome et al. 2022, DeMarino et al. 2022). The use of the particles has also been validated in several different types of samples including serum, plasma, CSF, urine, and saliva. This has now been explained in the methods section (lines 349-362)

There may be a typo: Line 51 – please define eCNS. (typo?)

We have corrected this typo on line 51.

Given that HIV and exosomes are similar in size, The authors should perform tests that discriminate virus from EV. Besides minor details, this work is very interesting and comparable to the EV field work.

We agree that it is important to discriminate between virus and EV. The individuals in the ALLHANDS cohort must meet strict inclusion criteria which includes viral suppression on antiretroviral therapy for greater than 1 year and continuous antiretroviral therapy. All participants have an undetectable viral load (unless otherwise noted in the patient characteristics table) and therefore the samples should be free of replicating and infectious particles. Nonetheless, we agree that it is critical to be able to isolate EVs in the absence of full viral suppression and have published methods for the purification of EVs away from virus using a series of isolation steps which effectively allow for isolation of a high yield EV prep in the absence of infectious, replicating virions (DeMarino et al. Antiretroviral Drugs Alter the Content of Extracellular Vesicles from HIV-1-Infected Cells. Sci Rep. 2018 and DeMarino et al. Purification of High Yield Extracellular Vesicle Preparations way from Virus. J Vis Exp. 2019).

Assuming the authors have only isolated EVs, the work is noteworthy, and the data analysis, interpretations, and conclusions seem correct. Viral RNA within EVs represents an exciting new prospect for how lentiviruses, such as HIV, disseminate and induce pathology.

We are thankful to the reviewer for this positive comment and we are grateful for the dedication of their time to review the manuscript.

Reviewer #3 (Remarks to the Author):

DeMarino et al. used ddPCR targeting different portions and forms of HIV transcripts in serum and CSF samples from a mixed cross-sectional/longitudinal cohort of people living with HIV to describe the link between neurocognitive dysfunction and HIV transcripts detection/levels in blood and CSF extracellular vesicles. The topic is of interest as the burden of HIV-associated neurocognitive disorders even when antiretroviral is successful remains a challenge in HIV infection care. The methodology is sound. The results presented here are relevant in the field, the manuscript is well written. However, I think some

information and analyses are missing, but can be further added, and the conclusions should sometimes be re-drawn, in regards to some points I mentioned below. Thus, I highlight here some major and minor points that I would like the authors to address.

Major Comments

A. My first point here is a reflection about what extracellular vesicles mirrors regarding the HIV reservoir. In other terms, do HIV transcripts detected in exosomes reflect the transcriptional activity of the viral reservoir? Would there be a possibility for a cell to transcribe HIV but not produce exosomes (or to to lesser a more important extent than compared to another cell)? Besides, extracellular vesicles (EV) can cross the blood brain barrier (BBB). So there might be a possibility that HIV transcripts detected in EV do not arise from the compartment in which they are detected. This could be shortly discussed to dampen some conclusions.

The reviewer poses an interesting point in asking to what extent do EVs mirror the HIV reservoir. In general, extracellular vesicles are representative of the cells from which they arise. However, the intricacies of viral RNA export into EVs for every cell type has not been fully elucidated. There is evidence suggesting the overall viral RNA profile in EVs corresponds to the transcript levels within the originating cells in myeloid lineage cells in the presence and absence of antiretroviral therapy (DeMarino et al. 2018) See lines 272-277. To fully address this question future studies should include postmortem brain tissue single cell sequencing paired with analysis of perimortem CSF and blood to fully address this point. The reviewer is correct, EVs have the potential to cross the BBB and there is a possibility that HIV transcripts detected in EVs do not arise from the compartment in which they are detected. We have included these critical points in the discussion section (lines 280-289).

B. Patient characteristics: It was rather difficult for me to understand if participants were treated (and the current drugs) or not and if they were in virological success. I could find some treatment information for the 43 longitudinal participants in extended table 1. But it's missing for the remaining 41 participants. I think this information should be added in Table 1. Same for the drugs, I think the numbers aren't quite right, it is stated that 6 participants are "currently on an integrase inhibitor", however when I recounted the number of participants treated with an integrase inhibitor in the extended table 1 for the 43 participants, at least 31 of them are currently on an integrase inhibitor. Please check all the numbers presented in this table. Besides the NNRTIs class and some NRTI drugs are missing in Table 1. Finally, viral load information is missing as well: are some of these participants viremic? This is really an important point that should be added for better understanding and interpreting the results presented here.

We are grateful to the reviewer for their careful review of the patient characteristics and pointing out this calculation. We have corrected the calculations in Table 1 and included ART information data for the entire cross-sectional cohort and longitudinal cohort for clarity in supplementary tables 1 and 2, respectively. We have also included reference numbering between the cross section and longitudinal cohorts for reader clarity. We have updated the plasma viral load information for all tested samples to aid in understanding and interpreting the presented results. The detection limit of our assay was 40 copies. In the cross-sectional cohort, all patients had undetectable viral loads except for one who had <200 copies. In the longitudinal cohort there was one other patient who at a single time point had 27,760 copies but was subsequently undetectable.

C. I think the author present very nice data in Fig1C-I, for which more analyses and conclusions could be drawn. First, a minor point for clarity: The authors mention that transcripts were not detected in all samples in Fig 1J but this does not appear in Fig1C-I. Are they plotted? If not please do so and place them at the limit of detection. If yes maybe plot them with an open circle or another symbol?

We agree with the reviewer in that it would be helpful to plot the undetectable patients in Fig 1. We have updated Fig 1C-I to include all individuals (not just individuals with detectable values) and updated the statistical tests to reflect the data points shown on the graph.

Second, even though correlation between both compartments in paired samples and for each transcript is described in the extended data. It would be nice to say whether it is the case or not in the results section.

We agree with the reviewer that it is important to report these findings in the results section. We have added a description of the relationship of the transcripts within each compartment and between compartments on lines 111-129 in the results section.

Third, the difference or absence of difference in HIV transcripts levels varies according to the transcript analyzed. Would the authors think that transcriptional blocks are different between blood and the CNS? I think it should be nice to assess that by performing paired comparisons between transcripts (to be performed separately for CSF and serum) like in the original Yukl paper. The authors mention these transcriptional blocks in the discussion section, but they did not perform statistical tests to prove it, but actually this seems quite feasible and will surely lead to some nice conclusions.

This is a great suggestion from the reviewer. The calculations are feasible with the approach we have taken to measure the transcripts within the CSF and serum. However, in the original Yukl paper, the calculations to determine transcriptional blocks at various stages of transcription are done by calculating a series of ratios utilizing TAR, Long LTR, PolyA, and Tat-Rev. The majority of the individuals tested in our study have no detectable copies of PolyA in the CSF or serum, likely due to the adequate viral suppression by ART drugs in this well followed cohort of individuals. This suggestion would be more feasible in a less well controlled cohort.

Fourth, Tat-Rev transcripts were observed to be at the lowest levels in the Yukl paper, here it seems at least in the CSF that Tat-Rev levels are as high as pol levels (in the CSF as well). I think it would be worth mentioning especially that the authors found an inverse correlation with the Psychomotor domain. This is quite interesting, and it would worth digging in the literature the implication of early and late HIV transcripts.

We greatly appreciated this suggestion. We have added a paragraph in the discussion (lines 254-265), "Remarkably, the multiply spliced Tat-Rev transcript was found in a high copy number relative to other transcript levels in the CSF which is contrast with the reported transcript levels in CD4+ T-cells from HIV patients in which Tat-Rev was found to be the lowest in abundance¹⁸. In that study, the patients were not fully controlled on ART, hence it is possible that complete control of viral replication by ART may result in increased spliced transcripts as shown previously (Johnson et al., PNAS 2013). Another possibility is that multiply spliced transcripts are actively transported out of the cell through a yet-to-be described mechanism. It may also suggest that the CNS has a relatively high number of 5'

defective proviruses which would be in line with recent reports which showed that defects in the major splicing donor (MSD) site caused a reduced efficiency in RNA dimerization and nucleocapsid binding which could allow for a shift towards a higher relative abundance of multiply spliced transcripts. This and the correlation of the multiply spliced transcripts with changes in the psychomotor domain require further study.”.

D. Some CSF HIV transcripts levels correlate with neurocognitive disorders scores. Do the authors have a hypothesis on why this is only the case for readthrough and long LTR? They mention the conservation of the LTR target sequence across HIV clades in the discussion section, so does this mean that for other transcripts with a more variable sequence there could have been an under detection thus leading to an absence of correlation with neurocognitive disorders scores?

The reviewer raises an interesting point about more variable sequences leading to an under detection of some transcripts in our assay. We agree that there is a potential for under detection of the transcripts with some of these primers sets. However, it is more likely that many of these coding regions have been deleted from the provirus altogether. There is literature to suggest that the proviral landscape changes over the duration of infection with the percentage of hypermutations declining over time and the percentage of large internal deletions increase over chronic infection (Pollack et al. 2017). Nonetheless, we feel it is an important caveat to include in our manuscript and we have included the potential for under detection of various transcripts in the discussion section lines 197-214.

E. Compartmentalization is hard to prove especially with few samples and few sequences retrieved. Besides, can we talk about compartmentalization in case EV containing HIV transcripts. The reservoir might be distributed similarly between CNS and blood but its transcriptional profile might be different. I would soften conclusions here. About the same intact sequences were found between serum and CSF, and the authors hypothesized seeding between those compartments. Could there be a passive transfer or is it local replication highly similar variants?

Agree. We have revised the sentence which now reads, “----suggesting the possibility of compartmentalization----” (line 252-253). This has also been revised in the abstract.

Minor Comments

1. Viral particles present in the blood (measured routinely for viral load assessments) were shown to arise mostly from the lymph node HIV-infected cells, would the authors think the serum EVs might come from there too? Is there a way to assess that maybe by phenotypic analysis of EVs?

This is an intriguing comment. It would be interesting to sequence the provirus from lymph node HIV-infected cells from an autopsy sample or biopsy as well as isolated extracellular vesicles and compare the viral sequences as a means of indirectly addressing this comment. Unfortunately, determining cell type origin is still an active area of investigation within the EV field and to date there is no gold standard to determine cell type origin of EVs. This has been discussed in lines 282-289.

2. Do the authors have an idea on HIV clade distribution across participants? In other terms might some

given clades be more frequently associated with HIV transcripts detection in CSF and/or neurocognitive disorders?

The cohort is comprised of several mixed clades and sequencing of all the individuals in the cohort and the relationship between proviral sequences, RNA transcript levels and neurocognitive function is an active area of investigation.

3. Fig3B-D Analyses were adjusted to viral load but what is the viral load (see my point above) are these individuals not supposed to be controlled? If so, why adjust to viral load? Is some of these individuals are viremic it would seriously alter the conclusions that were made.

This is an excellent point. Nearly all of the participants in the study have an undetectable viral load with rare exceptions. This information has been added to the patient characteristics tables (Extended data tables 1 and 2) for clarity. The model was adjusted to viral load to determine if those individuals represented outliers and should be excluded from the analysis (participants 11 and 15).

4. Serum readthrough seem to inversely correlate with CSF TAR, same for serum long LTR and CSF nef. Is there an hypothesis behind this?

We find this to be an interesting observation, but the relevance is unclear to us at this time. However the possibility that spliced transcripts are increased in the presence of ART has been discussed as mentioned above.

5. Some antiretroviral drugs successfully pass the BBB? Might drug levels in serum/CSF have a relation with HIV transcripts levels? If ART could influence transcription and in this case residual replication so maybe adjust to ART (as some drugs would better pass through the BBB than others) as well?

Yes, some antiretroviral drugs pass the blood brain barrier better than others. To address this comment, we have assigned each antiretroviral a CNS penetration effectiveness score which was originally published by Scott Letendre. The CPE for each drug was summed for individual and individuals were binned into either low or high CPE groups. A Wilcoxon rank sum test was used to assess significance to determine if CNS penetrance affected viral transcript levels in the CSF and serum. We included this data in the Extended Data Figure 3 and included the results in the manuscript. This suggestion has generated data that helps to underscore the importance of the need for novel HIV therapeutics which target viral transcription.

6. I think it would be nice for the readers to better plot the longitudinal data: maybe the time since undetectability in x axis and the HIV transcript levels in y axis and link points from a given participant to see individual trajectories. Thus we could see what are the trends: some patient increase their levels, some not.

We have plotted the longitudinal data as suggested. These plots can be found in Extended Data Fig 4. We have included a general description of the overall trends or lack thereof in the results section of the manuscript.

Reviewers' Comments:

Reviewer #1:

Remarks to the Author:

The authors have responded adequately to the critiques, thereby strengthening the study even more !

Reviewer #2:

Remarks to the Author:

The manuscript is well-written, and the findings are intriguing. Researchers demonstrated that despite viral suppression, HIV RNAs are still detected in EVs of aviremic HIV patients. Overall, correlating EV RNA content with neurocognitive impairment indices and evaluating the distribution of various HIV RNAs in EVs is novel and intriguing.

The authors addressed all previous issues in this revised manuscript except the one listed below.

Minor issue:

(i)Although the EVs were quantified, characterization of the EVs (confirmation of size or surface markers, i.e., CD63, ALIX, TSG101, etc.)is lacking.

Dynamic light scattering, transmission electron microscopy, NTA, etc., could have been performed to determine which EV populations were isolated from the specimens. Essentially, the Nanotrap isolations should be authenticated and shown in the extended data (if feasible)

Despite a minor weakness, this manuscript described exciting findings with extracellular vesicle content and HIV RNA and should undoubtedly be published.

Reviewer #3:

Remarks to the Author:

The authors have addressed all my comments and concerns, by adding new analyses to their data and presenting new graphs that provide interesting insights to the field. I feel the manuscript has been greatly improved.

I would like to thank them for their answers and comments.

REVIEWERS' COMMENTS

Reviewer #1 (Remarks to the Author):

The authors have responded adequately to the critiques, thereby strengthening the study even more!

We thank this reviewer for the time and energy they devoted to reviewing this manuscript.

Reviewer #2 (Remarks to the Author):

The manuscript is well-written, and the findings are intriguing. Researchers demonstrated that despite viral suppression, HIV RNAs are still detected in EVs of aviremic HIV patients. Overall, correlating EV RNA content with neurocognitive impairment indices and evaluating the distribution of various HIV RNAs in EVs is novel and intriguing.

The authors addressed all previous issues in this revised manuscript except the one listed below.

Minor issue:

(i) Although the EVs were quantified, characterization of the EVs (confirmation of size or surface markers, i.e., CD63, ALIX, TSG101, etc.) is lacking.

Dynamic light scattering, transmission electron microscopy, NTA, etc., could have been performed to determine which EV populations were isolated from the specimens. Essentially, the Nanotrap isolations should be authenticated and shown in the extended data (if feasible)

Despite a minor weakness, this manuscript described exciting findings with extracellular vesicle content and HIV RNA and should undoubtedly be published.

We agree that validation of methods is essential for all research. This particular method has been validated in several studies (Jaworski et al. 2014, Sampey et al. 2016, Barclay et al 2017, DeMarino et al. 2018, Anderson et al. 2018, Barclay et al. 2018, Pinto et al. 2019, DeMarino et al. 2019, Henderson et al. 2019, Branscome et al. 2022, DeMarino et al. 2022), many of which the first author of this manuscript has written. These particles have been validated in all of the sample types used in this study and therefore is a sound approach to the bulk isolation of extracellular vesicles.

Reviewer #3 (Remarks to the Author):

The authors have addressed all my comments and concerns, by adding new analyses to their data and presenting new graphs that provide interesting insights to the field. I feel the manuscript has been greatly improved.

I would like to thank them for their answers and comments.

We thank the reviewer for the time and energy they devoted to reviewing this manuscript.